# Potential of a Non-Contrast-Enhanced Abbreviated MRI Screening Protocol (NC-AMRI) in High-Risk Patients under Surveillance for HCC

**DOI:** 10.3390/cancers14163961

**Published:** 2022-08-17

**Authors:** François Willemssen, Quido de Lussanet de la Sablonière, Daniel Bos, Jan IJzermans, Robert De Man, Roy Dwarkasing

**Affiliations:** Department of Radiology and Nuclear Medicine, Erasmus University Medical Center, ’s Gravendijkwal 230, 3015 CE Rotterdam, The Netherlands

**Keywords:** hepatocellular carcinoma, non-contrast-enhanced abbreviated MRI, surveillance, double reading, training session

## Abstract

**Simple Summary:**

According to guidelines from the European Association for the Study of the Liver (EASL) and American Association for the Study of Liver Diseases (AASLD), abdominal ultrasound (US) is recommended for surveillance of hepatocellular carcinoma (HCC) in high-risk patients. However, US is limited as a surveillance modality for various reasons. Magnetic resonance imaging (MRI) is generally considered a better modality for detection of early HCC, but too elaborate in a surveillance setting. Consequently, abbreviated MRI (AMRI) protocols are investigated for surveillance purposes. The aim of our study was to evaluate the potential of non-contrast-enhanced AMRI (NC-AMRI) for surveillance of HCC, using multiple readers to investigate inter-observer agreement and the added value of double reading. We found that NC-AMRI presents a valuable screening tool for HCC and that double reading improves the sensitivity and specificity of HCC detection.

**Abstract:**

Purpose: To evaluate NC-AMRI for the detection of HCC in high-risk patients. Methods: Patients who underwent yearly contrast-enhanced MRI (i.e., full MRI protocol) of the liver were included retrospectively. For all patients, the sequences that constitute the NC-AMRI protocol, namely diffusion-weighted imaging (DWI), T2-weighted (T2W) imaging with fat saturation, and T1-weighted (T1W) in-phase and opposed-phase imaging, were extracted, anonymized, and uploaded to a separate research server and reviewed independently by three radiologists with different levels of experience. Reader I and III held a mutual training session. Levels of suspicion of HCC per patient were compared and the sensitivity, specificity, and area under the curve (AUC) using the Mann–Whitney U test were calculated. The reference standard was a final diagnosis based on full liver MRI and clinical follow-up information. Results: Two-hundred-and-fifteen patients were included, 36 (16.7%) had HCC and 179 (83.3%) did not. The level of agreement between readers was reasonable to good and concordant with the level of expertise and participation in a mutual training session. Receiver operating characteristics (ROC) analysis showed relatively high AUC values (range 0.89–0.94). Double reading showed increased sensitivity of 97.2% and specificity of 87.2% compared with individual results (sensitivity 80.1%–91.7%–97.2%; specificity 91.1%–72.1%–82.1%). Only one HCC (2.8%) was missed by all readers. Conclusion: NC-AMRI presents a good potential surveillance imaging tool for the detection of HCC in high-risk patients. The best results are achieved with two observers after a mutual training session.

## 1. Introduction

HCC is the most frequent primary tumor of the liver and the third most common cause of cancer-related deaths annually worldwide [1]. The majority of HCCs occur in patients with viral hepatitis B and C, non-alcoholic fatty liver disease (NAFLD), non-alcoholic steatohepatitis (NASH), and liver cirrhosis. These patients are thus considered as high-risk patients for developing HCC [2]. Surveillance with imaging is recommended in high-risk patients for timely detection of early HCC, which may lead to curative treatment [3]. 

Current surveillance guidelines of both the EASL and AASLD recommend bi-annual US for surveillance [4,5]. Advantages of US include its wide availability and low costs. However, US has poor sensitivity for detection of early HCC, especially in cirrhotic livers [6]. A recent meta-analysis concluded that US has a sensitivity of only 47% for the detection of early HCC in patients with cirrhosis [7]. In addition, US proved technically inadequate to “penetrate” cirrhotic livers and rule out HCC in more than 20% of patients [8]. In our hospital, a tertiary referral center for liver disease, patients who are clinically considered to be unsuited for screening with US are screened yearly with full liver CE-MRI. 

MRI and computed tomography (CT) are considered to be superior over US for detection and diagnosis of HCC, particularly in patients clinically considered to be unsuited for US owing to obesity, liver steatosis, fibrosis, or cirrhosis [6,9]. Although CT may be cost-effective for HCC screening [10], it is considered to be unfavorable for surveillance purposes because of repetitive and thus cumulative radiation exposure. As imaging for HCC requires at least three phases, i.e., arterial, portal-venous, and delayed phases after intravenous contrast administration, the radiation dose is estimated to be around 20 mSv per examination [11]. Although, a recent meta-analysis found that a cumulative dose of 100 mSv and probably 200 mSv may not increase the risk of carcinogenesis for patients; this cumulative dose is reached within 5 years in the case of HCC surveillance [12]. As HCC surveillance is a lifelong activity, the cumulative dose may far exceed 200 mSv. MRI seems promising; however, higher costs, the use of intravenous contrast agents, and longer examination time are drawbacks that make MRI unattractive for surveillance purposes. These drawbacks may be overcome with AMRI liver screening protocols that preserve good sensitivity for the detection of HCC. 

Recently, different AMRI protocols have been proposed for surveillance purposes with promising results [13,14]. In general, AMRI protocols can be categorized as protocols using intravenous contrast agents, namely CE-AMRI, or protocols without contrast material, namely NC-AMRI. Our proposed surveillance protocol is an NC-AMRI protocol, which consists of three sequences in the axial plane: DWI, T2W imaging with fat saturation, and T1W in-phase and opposed-phase imaging. These sequences are readily part of the recommended standard clinically applied full MRI protocol for imaging and diagnosis of focal liver lesions [15]. This reduced combination of sequences favorably reduces the total acquisition time to just 12.5 min, including localizer and calibration images. 

Efforts to further improve screening results that are nowadays common practice in breast cancer screening, for example, have to the best of our knowledge not been explored in HCC screening. Potential improvements (e.g., reduce false-positive recall rates and increase cancer detection rates) are supplemental training to readers and double-reading [16,17]. 

The purpose of this retrospective study was to evaluate NC-AMRI for HCC detection in high-risk patients who underwent yearly full MRI for surveillance of HCC. 

## 2. Materials and Methods

The medical ethical committee of our institution granted permission for this retrospective study and informed consent was waived, as the study was performed with anonymized data and in accordance with the Central Committee on Research involving Human Subjects. 

### 2.1. Patient Selection

Consecutive patients who underwent MRI of the liver for screening of HCC in a surveillance program between January 2010 and January 2019 were reviewed. Patients received yearly full MRI protocol, owing to failed surveillance with abdominal US, mostly because of fatty infiltration of the liver, advanced liver cirrhosis, or obesity. For inclusion, patients received at least two full MRI examinations. In patients without HCC (i.e., HCC naïve), the second-to-last MRI was included for further analysis. The reason for this was that the last MRI was considered necessary as a reference to ensure that the patient was truly HCC naïve. In the event of HCC, the MRI examination with first detection of HCC was included for analysis.

Patients’ demographics were retrieved from medical records. For each patient, gender, age, and presence of liver disease with underlying cause were registered. The number of patients with and without HCC was recorded. The Li-RADS classification of focal liver lesions per patient was documented. The Child–Pugh classification and clinical stage of patients with HCC according to the Barcelona Clinic Liver Cancer system (BCLC) and the treatment were noted.

### 2.2. Full MRI Liver Protocol and Non-Contrast-Enhanced Abbreviated MRI Surveillance Protocols

The MRI examination for all patients was performed on a 1.5 Tesla system (Signa, GE Healthcare) with use of a dedicated 8–16-channel abdominal coil. This protocol included the following sequences: T2W fast spin echo, in axial and coronal plane; axial T1W gradient-echo sequences in-phase and opposed-phase; axial DWI with at least three b-values; axial T2W imaging with fat saturation; and T1W dynamic CE series, with arterial phase acquired using bolus triggering, repeated at least four times after injection of 7.5 cc Gadobuterol 1 mmol/mL (Bayer Schering, Leverkusen, Germany) at a rate of 2 mL/sec and saline flush of 30 mL, followed by delayed coronal 3D T1W images. Total imaging time, including localizer and calibration, was around 27 min. Details of these sequences are shown in Table 1.

Our NC-AMRI protocol consists of the above-mentioned axial T1W in-phase and opposed-phase, DWI, and T2W with fat-saturation sequences, with a total acquisition time of 12.5 min (Table 1). Further reductions in total patient handling time owing to the omission intravenous line placement for CE acquisitions were not measured. An expected advantage of a short duration session is that the MRI examination is better tolerated and potentially results in better patient compliance as well as less artifacts and/or missing sequences. Because we do not know the true effect of the abbreviated protocol on image quality (possible confounder), we included only those patients of the study population who had full execution of all three MRI sequences that constituted the NC-AMRI protocol.

### 2.3. Image Interpretation of the NC-AMRI Protocol

Only NC-AMRI sequences per MRI examination of the study population were extracted, anonymized, and uploaded onto a separate research server. An open-source imaging informatics platform (XNAT version 1.7, Buckner Lab, Washington University School of Medicine, St. Louis, MO, USA) [18] was used for this purpose, which is a secured platform and accepted by the Medical Ethical Committee to achieve anonymity, according to the privacy guidelines.

Three readers with different levels of experience evaluated the NC-AMRI sequences of each patient separately and were blinded for the remaining sequences of the full MRI examination, including all previous imaging studies and clinical information. Readers I and II were abdominal radiologists with fifteen and twelve years, respectively, of professional experience in liver imaging at a tertiary center for hepatobiliary diseases. Reader III was a radiologist with six years of professional experience in general and abdominal radiology at a middle-sized primary hospital. Experienced reader I and less-experienced reader III held a joint training session prior to scoring. Experienced reader II purposefully did not participate in this joint training session. The joint training session was conducted by reviewing ten patients with liver cirrhosis from a teaching file collected by experienced reader I, including five cases with HCC as well as five cases with benign and challenging non-HCC cases such as confluent fibrosis. None of these teaching cases were included in this study. T2W imaging with fat saturation and DWI (restricted diffusion) typically depict HCC as a lesion with a high signal intensity (SI), while T1W in-phase and opposed-phase imaging may be used to confirm the presence of the lesion and to detect intralesional fatty deposits, which are oftentimes seen in early HCC or well-differentiated HCC [19,20].

The readers considered image quality, presence of focal lesions, size, segment location, visibility on the different sequences, and conclusion of a benign observation or possible HCC. Data were registered, digitally secured, and stored using a standardized anonymized and coded clinical reporting form in the online clinical software program OpenClinica (version 3.1.3.1, OpenClinica, Waltham, MA, USA) [21]. OpenClinica is an open-source electronic case report form, developed for clinical data of trials and recommended by the institutional medical ethical committee. Each patient case was consequently graded on a five-point scale: 1 = no lesion, 2 = benign lesion, 3 = possible HCC, 4 = probable HCC, and 5 = confident HCC. Using this five-point scale may indicate whether ROC curve analysis with score of 3 to 5 (possible, probable, and confident HCC) would justly qualify for further evaluation with full MRI liver protocol to confirm (or exclude) HCC. The scoring was case-based, i.e., patient-based, not lesion-based, as the primary goal was to correctly differentiate between high-risk patients that necessitate additional full liver MRI for confirmative diagnosis from those patients who do not need further examination and can remain under regular surveillance. The scoring results were first evaluated for each reader separately and then for all readers combined based on majority votes to simulate double-reading.

### 2.4. Reference Standard

The reference standard for HCC was based on the corresponding full MRI protocol with confirmation by the multidisciplinary liver tumor board (MDTB). The institutional MDTB comprises specialized radiologists, hepatologists, surgeons, radiotherapists, and oncologists. The reference standard for benign lesions, including technical artifacts, was based on follow-up full MRI reports. 

Final diagnosis of all focal lesions (benign and HCC) was registered with the Li-RADS V. 2018 [22] classification system and validated with clinical follow-up information. 

### 2.5. Data Analysis and Statistical Methods

Descriptive statistics were used to describe the study population. The primary analysis was patient-based. The difference between categorical variables was presented by numbers and percentages and tested with the Fisher exact test. The conclusiveness of the findings from each reader was presented as numbers and percentages. For further analysis of readers’ scoring data, the qualification of confident, probable, or possible HCC was considered positive for HCC. Sensitivity, specificity, ROC/AUC value, positive predictive value (PPV), negative predictive value (NPV), and accuracy of NC-AMRI were calculated using SPSS software (version 21, IBM) for each reader separately and for all readers together using the majority of votes. AUC of 1.0 indicated a perfect model, whereas 0.5 indicated a very poor model. Generally, AUC > 0.7 indicates a good model [23]. Inter observer agreement was calculated using Cohen’s kappa statistics, to be interpreted as follows: values 0.00–0.20 none to slight, 0.21–0.40 fair, 0.41–0.60 moderate, 0.61–0.80 substantial, and 0.81–1.00 almost perfect agreement. Interclass correlation coefficients, using the two-way random effects model, with 95% confidence interval were calculated using the Fisher test [24]. Mann–Whitney U test was used to evaluate the differences in variables with a continuous distribution across categories. The association among categorical variables was assessed by Chi-squared test or Fisher’s exact test. All the tests were two-sided, and *p*-values of <0.05 were considered statistically significant. 

### 2.6. Review by Consensus after Data Analysis

After completion of data analyses, the original imaging data and clinical information were made available for all three readers, for selecting illustrative cases and figures for this publication. 

## 3. Results

A total of 240 consecutive patients were eligible for inclusion. Twenty-five patients were excluded because of substandard quality MRI. These patients lacked one sequence of our NC-AMRI protocol (4 patients) or had too disturbing motion artefacts of the NC-AMRI sequences (21 patients). Minor and moderate motion artefacts were accepted. The remaining 215 patients, 149 (69.3%) male and 66 (30.7%) female with a mean age of 56 years (range 19–81 years), were included for final analysis. The study population consists of 179 (83.3%) HCC-naïve patients and 36 (16.7%) patients with HCC. Most patients in the HCC-naïve subgroup had cirrhosis (143/179, 79.9%) followed by non-cirrhotic chronic hepatitis B (30/179, 16.8%) and non-cirrhotic hepatitis C (4/179, 2.4%). In the HCC subgroup, most patients (91.7%, 33/36) had cirrhosis and only one patient (2.8%) had chronic hepatitis B. 

Twenty-eight patients with HCC had Li-RADS 5 lesions (77.8%) and 8 patients had Li-RADS 4 lesion (22.2%). HCC-naïve patients had no focal lesions in 55 patients (30.7%); Li-RADS 1 and 2 lesions in 95 patients (53.1%), and Li-RADS 3 lesions in 29 patients (16.2%) (Figure 1). Of the 36 patients with HCC, 30 (83.3%) had Child–Pugh score A, and 6 patients (16.7%) Child–Pugh score B. BCLC stage was 0/A in 25 patients (69.4%), B in 6 patients (16.7%), and C in 4 patients (11.1%). One patient was treated in another hospital (2.8%). The treatment patients underwent was radiofrequency ablation in 16 patients (44.4%), followed by liver transplantation in 2 patients (12.5%). Eight patients underwent resection (22.2%) and 6 underwent patients trans arterial chemo-embolisation (16.7%), followed by liver transplantation in 2 patients (33.3%). Four patients received chemotherapy (11.1%) and 1 patient received a liver transplantation immediately (2.8%), and 1 patient was treated elsewhere (2.8%).

### 3.1. Cases with HCC

The mean HCC lesion size was 31 mm, including seven patients with more than one HCC lesion. Readers I and III, who held a training session prior to scoring, had the highest sensitivity per patient of 97.2% (35/36) and 91.7% (33/36), respectively, as compared with reader II, with 80.1% (29/36) sensitivity. Conversely, reader II had a much greater specificity of 91.1% (163/179) than reader I with 82.1% (147/179) and reader III with 72.1% (129/179) (Table 2). Nonetheless, there was a substantial agreement between the most experienced readers I and II (k = 0.64), whereas the agreement of the less experienced reader III with both experienced readers I and II was only moderate (k = 0.55 and 0.49, respectively) (Figure 2). ROC analysis showed a good AUC for all three readers (reader I 0.94, reader II 0.88, and reader III 0.93) (Figure 1). 

Simulations for double-reading showed a good interclass correlation coefficient of all readers (0.88; 95% CI 0.843–0.906). Using majority vote, sensitivity and specificity increase to 97.2% (35/36) and 87.2% (156/179), respectively, with NPV ranging from 95.9% to 99.4%. HCC was correctly detected and interpreted by all three readers in 61.1% of patients (22/36) (Figure 2). In the remaining HCC patients (36.1%, 13/36), any two readers interpreted the lesions as confident, possible, or probable HCC. In only one patient (2.8%) was the HCC lesion missed by all readers. Upon review by consensus after data analysis, the readers agreed that the solitary lesion in segment 3 was not discernable on T2W FS and merely very slightly hyperintense on DWI and T1W images, and that the lesion was missed in this patient on NC-AMRI owing to a sub-optimal image quality of the left liver lobe with an overall lack of signal intensity on DWI (Figure 3). Consequently, confident, probable, and possible HCC was interpreted correctly in all, but one patient using majority vote.

Area under the curve (AUC): reader I 0.94 (95% CI 0.90–0.99), reader II 0.88 (95% CI 0.81–0.96), and reader III 0.93 (95% CI 0.89–0.97). AUC of 1.0 indicated a perfect model, whereas 0.5 indicated a very poor model. Generally, AUC > 0.7 indicates a good model.

### 3.2. HCC-Naïve Cases

In HCC-naïve cases, diagnosis was correct in 87.2% (156/179) using majority vote. All three readers correctly scored 65.4% (117/179) of the patients as negative for HCC. In about one-eighth of the patients (12.8%; 23/179), two readers qualified a benign lesion as positive (i.e., possible, probable, or confident) for HCC, and about one-fifth of the patients with benign lesions (21.8%; 39/179), while one reader scored a benign lesion as positive for HCC (Figure 2). Examples of such ‘false positive’ lesions are (atypical) hemangiomas that are for example just mildly hyperintense on T2W images (Figure 4). 

## 4. Discussion

Our retrospective study based on a simulated NC-MRI liver protocol in a surveillance population shows promising sensitivities (range 80–97%) and specificities (range 72–91%) for both highly experienced and less experienced abdominal radiologists. Although we did not perform a direct comparison with US in our study population, the results hold promise that NC-AMRI may be a substantial improvement over US (reported sensitivity of 47%) for the detection of lesions that resemble HCC and warrant further evaluation with full MRI for confirmative final diagnosis [7,25]. 

Because of disappointing reports on US surveillance for HCC in high-risk patients, surveillance with AMRI gained much attention in recent years [3,4,5,6,7,9,13,14,19,20,26,27,28,29]. In a recent meta-analysis, sensitivity and specificity for both CE-AMRI and NC-AMRI protocols were comparably good [29]. The reported sensitivity and specificity were 86% and 94%, respectively, for NC-AMRI and 87% and 94%, respectively, for CE-AMRI. NC-AMRI with or without in- and opposed-phase T1W imaging had non-significant differences in sensitivity and specificity [19]. In another meta-analysis, comparing hepatobiliary phase CE-AMRI and NC-MRI for detection of HCC, sensitivity was higher in favor of CE-MRI over NC-AMRI, but specificity was lower (sensitivity 87% vs. 82% and specificity 93% vs. 98%, respectively) [30]. However, some of the included studies in this review were not performed in a surveillance setting for HCC detection. In spite of the fact that some limited differences in sensitivity and specificity between different AMRI protocols for surveillance of HCC were reported, the sensitivity and specificity are generally much better compared with US [3,7,9,13,19,25,26,27,29,30,31]. Furthermore, our results concur with these reports. Although opinions may differ, we propose that, ideally, an AMRI imaging surveillance protocol should be non-invasive (i.e., no intravenous contrast agent administration) with an examination time comparable to that of US (ten to fifteen minutes), but with improved detection capabilities of HCC compared with US. Concerns on the possible long-term risks with repeated use of gadolinium-based MRI contrast agents in a surveillance setting, with possible accumulation of Gadolinium in the brain, combined with the added benefit of non-invasiveness and lower costs, would thus make NC-AMRI the preferred choice over CE-AMRI protocols [32,33,34]. Possible further reductions in patient handling time and/or costs resulting from the omission of intravenous contrast-agent injection from the MRI screening protocol are beyond the scope of this study. Nonetheless, patient-in-room time of our proposed NC-AMRI protocol is relatively comparable to the time needed for an US examination. Moreover, with optimization of the work flow in the MRI unit and faster sequences, especially T2W imaging, which is under development, imaging time may further be reduced [35]. In addition, other studies with even fewer AMRI sequences (e.g., DWI- only, or DWI with T2W) to further reduce imaging time may have a better sensitivity than US [20,25]. Still, even in a screening setting, not only a good sensitivity, but also a good specificity is warranted. 

In our study, the readers had no access to prior examinations, which would reflect a (most pessimistic) screening situation in which all patients are considered newly enrolled. For this reason, it is expected that the relatively high false-positive detection rate of 13% will likewise be lower in daily practice when the readers have access to prior studies for a fair comparison. Previously established benign lesions such as (atypical) hemangiomas as illustrated in Figure 4 can then be verified as such. Nonetheless, we wish to emphasize that, with screening for HCC, all doubtful lesions would qualify for confirmative diagnosis with full MRI liver protocol. After confirmation of the benign nature (Li-RADS 1 and 2) of the lesions, the patient may proceed with the regular surveillance schedule using NC-AMRI. 

It is possible that early HCC, which are only clearly seen with full MRI, may be missed using NC-AMRI. In our study, there was one such lesion that was consequently missed by all three readers (LR5, size < 2 cm), as illustrated in Figure 3. Theoretically, these lesions could be detected on the following surveillance when signal intensity changes may develop on the NC-AMRI sequences. Future studies are warranted to reveal the true incidence of early HCCs that are only seen with contrast-enhanced series without accompanying signal intensity changes on T2W, T1W, and/or DWI. Furthermore, future studies should provide us with information on the occurrence of hypovascular early HCC, which may be not apparent on contrast-enhanced series, but can be recognized on NC-AMRI. These are important considerations for embracing AMRI protocols for HCC surveillance. 

The further improved sensitivity (97%) and specificity (87%) of NC-AMRI based on majority vote suggests that double-reading may be preferable over single-reading in HCC imaging surveillance programs. These improvements appear in line with increases in sensitivity of more than 10% when double-reading was applied in an imaging surveillance program for breast cancer [36,37]. To the best of our knowledge, no other studies addressed this point in HCC imaging surveillance. Future prospective studies may show whether double-reading is indeed preferable over single reading in imaging surveillance programs for HCC. Possible variations in double-reading (e.g., consensus reading to resolve discordant results, or third reader arbitration) or future single reading combined with computer-aided detection are beyond the scope of this study. In addition, the extra personnel and time required for double reading needs further investigation, especially when considering the cost-effectiveness analysis of NC-AMRI protocols compared with CE-AMRI protocols and US screening.

Inter-observer agreement was substantial for the experienced readers and, in accordance with a recent systematic review and meta-analysis, inter-observer variability of 0.72 (95% CI 0.62–0.82) for NC-AMRI for the detection of HCC was determined [38]. However, the moderate agreement in the combination with the less-experienced reader was not determined. On the other hand, AUC was good for all readers. It is likely that the joint training session held by experienced reader I and less-experienced reader III prior to scoring contributed to the results of the less-experienced reader in particular. This is in line with reports on the utility of supplemental training to improve radiologist performance in breast cancer screening [16,17]. Positive effects of the joint training session prior to screening may also be reflected by the comparably high sensitivities of readers I and III, with respect to experienced reader II, who on purpose did not participate in the prior joint training session. For screening studies, a high sensitivity yield is most favorable. Conversely, the less-experienced reader demonstrated lower specificity and higher false-positive detection rates compared with the experienced readers. False-positive detection rates should be at an acceptable level to prevent redundant referrals for full CE-MRI and could add to increased costs and patient anxiety. Future studies may indicate the level of experience and supplemental training required for HCC surveillance with NC-AMRI. 

Another point to consider is that all patients were imaged using a 1.5T MRI system, and no use was made of 3T systems that might provide even better lesion detection owing to a higher signal-to-noise ratio. In our opinion, 1.5T is more widely available and perhaps more robust than 3T, as it is less susceptible to motion artifacts that might obscure parts of the liver, especially of the sub-capsular regions and lateral left liver lobe [39]. It is debatable whether the one missed HCC in our study in segment 3 (Figure 3) would have improved visibility on NC-AMRI on 3T. Nonetheless, our results with 1.5T MRI are in line with studies using 3T screening MRI [40]. This may imply that screening with MRI can be performed with both 1.5 and 3.0T MRI systems. 

Our study has some limitations. Foremost are the retrospective nature and simulated analysis of our NC-AMRI. However, our results are in line with other publications on NC-AMRI protocols for HCC detection in a diagnostic cohort [19]. Additionally, the exclusion patients with missing sequences and severe artefacts may have affected our results, as addressed in the Materials and Methods section. Future studies may show whether a shortened duration MRI examination is indeed better tolerated by the patient, resulting in better patient compliance, with reduced artifacts and/or missing sequences. Another limitation might be that not all lesions were confirmed with histopathology examination. We did, however, classify that HCC lesions conform to the Li-RADS system and, inherent to Li-RADS, no histopathologic examination is needed for Li-RADS 5 lesions. Still, eight Li-RADS 4 lesions were classified on NC-AMRI as possible HCC. These Li-RADS 4 lesions were discussed within the MDTB and accepted as HCC including treatment with local ablative therapy. Li-RADS 4 lesions could theoretically imply that these were not HCC. Nonetheless, it was reported that 74% of Li-RADS 4 lesions are indeed HCC and the decision for instant treatment of these lesions is best discussed in the MDTB [41,42]. Furthermore, differentiation between benign lesions like hepatic hemangioma and HCC is difficult, and mostly requires contrast administration [15]. Lastly, our study did not address the appropriate time interval between imaging surveillance sessions, and one might also argue that MRI ideally be performed every six months, as recommended for US screening. Unfortunately, that was not possible in this retrospective inclusion. Our cohort does, however, reflect real-life practice in a liver expertise center with multidisciplinary clinical decision making. As evidence increases, we have come to a crossroad of embracing AMRI as the standard for imaging surveillance of early HCC in a high-risk population. Assessment of cost-effectiveness and agreements for reimbursement are the next challenges to overcome. There are promising reports that AMRI may be cost-effective for HCC surveillance in some countries. However, this may not be the case for all countries, but dependent on health-care reimbursement regulation [10,32] and other standards (e.g., single- versus double-reading, or screening in primary hospitals versus tertiary referral centers). 

## 5. Conclusions

Our study demonstrates that NC-AMRI of the liver may present a valuable surveillance modality for HCC surveillance in high-risk patients. Diagnostic accuracy may further be improved with double-reading. In addition, implementing a mutual training session is promising, and the exact role of this needs further evaluation. 

## Data Availability

The data presented in this study are available on request from the corresponding author. The data are not publicly available owing to storage on a local server.

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
