# Peer review of "Potential of a Non-Contrast-Enhanced Abbreviated MRI Screening Protocol (NC-AMRI) in High-Risk Patients under Surveillance for HCC"

_cancers, 2022, doi:10.3390/cancers14163961_

Round 1

Reviewer 1 Report

Comments to Author: The manuscript has been revised well.

Author Response

Thank you for reviewing our article and your comments and suggestions. We are glad we were able to improve our article thanks to your suggestions.

Kind regards,

Reviewer 2 Report

This study aimed to evaluate non-contrast-enhanced abbreviated MRI (NC-AMRI) for the detection of HCC in high-risk patients. This manuscript is well written, but there are some issues that need to be addressed before publication. Considering that many studies have already been published on the accuracy of AMRI (e.g., reference 24), a more rigorous analysis is required in this study.

1. Detection of “early-stage” HCC amenable to curative treatment through surveillance test is very important because any but the earliest stages of HCC is usually incurable. In this context, data on the clinical staging of patients with HCC (i.e., Barcelona Clinic Liver Cancer system) detected by surveillance should be provided in this study. Also, please indicate how many of the included patients ultimately received curative treatment.

2. Please compare the diagnostic accuracy of NC-AMRI with that of full liver CE-MRI for detecting HCC. In addition, it would be better to compare the diagnostic accuracy between different NC-AMRI protocols (DWI + T2W + T1W in-phase and opposed-phase imaging versus DWI + T2W imaging).

3. Please provide additional data on baseline characteristics of enrolled patients such as laboratory findings (including tumor markers) and Child-Pugh score.

4. The imaging findings of typical HCC found in NC-AMRI should be described in the M & M section. Also, I wonder if nodules smaller than 1 cm were considered positive.

5. In the M & M section, it is mentioned that an annual liver MRI was performed on patients deemed unsuitable for screening with US examination. Please describe in detail the cases in which the US exam was unsuitable. For example, were they severely obese or patients with advanced cirrhosis?

6. The results of this study do not sufficiently support the following conclusion: “Best results are achieved with two observers after a mutual training session.” A training session was held only between readers I and III, but there reported NC-MRI diagnostic accuracy does not appear to be significantly different from that of reader II. Furthermore, even after the training session, interobserver agreement between readers I and III was not good (kappa, 0.56). Therefore, it is necessary to revise the conclusion more conservatively.

7. Tables 2 and 5 should be considered figures.

End of comments.

Author Response

Thank you for reviewing our article and your comments and suggestions. In the attachment you will find our response in italics.

Kind regards,

Reviewer 3 Report

This is a nicely performed study. 

I have the following comments that the authors should address

 In M&M section

1. The authors should clearly indicate which ICC they used because they are at least three different ICCs and add the following corresponding reference

Benchoufi M, Matzner-Lober E, Molinari N, et al. Interobserver agreement issues in radiology.  Diagn Interv Imaging 2020;101(10):639-641. doi: 10.1016/j.diii.2020.09.001.

2. The authors must report ICC with their 95% confidence intervals and add the following corresponding reference

Benchoufi M, Matzner-Lober E, Molinari N, et al. Interobserver agreement issues in radiology.  Diagn Interv Imaging 2020;101(10):639-641. doi: 10.1016/j.diii.2020.09.001.

In Results

3. The authors should add the 95% CI of kappa values and AUC.

4. This comment also applies to sensitivity, specificity, accuracy, NPV and PPV.

5. In addition, sensitivity, specificity, accuracy, NPV and PPV should not be reported as percentages only but the authors should add the corresponding proportions used to calculate percentages per common scientific publication guidelines

Discussion

6. The authors should acknowledge as a limitation of their study that the use of contrast material is needed for differentiating between HCC and other benign conditions

7. The authors should add the following reference to support this

Tan Y, Xie XY, Li XJ, et al. Comparison of hepatic epithelioid angiomyolipoma and non-hepatitis B, non-hepatitis C hepatocellular carcinoma on contrast-enhanced ultrasound. Diagn Interv Imaging 2020;101(11):733-738.

Tables

8. The authors must use dots and not commas for decimals in Table 4. 

Figures

9. On Figure 1, decimals should read 0.0, 0.2 and not 0,0, 0,2 and so on

 References. The authors should add the following recent references about abbreviated protocols to enrich their discussion

10. Khatri G, Pedrosa I, Ananthakrishnan L, et al Abbreviated-protocol screening MRI vs. complete-protocol diagnostic MRI for detection of hepatocellular carcinoma in patients with cirrhosis: An equivalence study using LI-RADS v2018. J Magn Reson Imaging 2020;51(2):415-425. doi: 10.1002/jmri.26835

11. Chan MV, McDonald SJ, Ong YY, et al. HCC screening: assessment of an abbreviated non-contrast MRI protocol. Eur Radiol Exp 2019;3(1):49. doi: 10.1186/s41747-019-0126-1

12. Park SH, Kim B, Kim SY, et al. Abbreviated MRI with optional multiphasic CT as an alternative to full-sequence MRI: LI-RADS validation in a HCC-screening cohort. Eur Radiol 2020;30(4):2302-2311. doi: 10.1007/s00330-019-06546-5.

Author Response

(The authors gave the same response as above.)

Round 2

Reviewer 2 Report

The revision of the manuscript was done satisfactorily. I appreciate the efforts of the authors to make this change.

I would like to point out one more thing in the Discussion section (second paragraph). You mentioned that the diagnostic accuracy of the CE-AMRI and NC-AMRI protocols was comparable. However, another recent meta-analysis reported that HBP CE-AMRI showed significantly higher sensitivities for detecting HCC than NC-AMRI, but significantly lower specificities (Cancers (Basel). 2021;13(12):2975). Please briefly discuss in the manuscript that there have been different results in terms of comparison between AMRI protocols.

End of comments.

Author Response

Thank you for the comment, we rephrased the paragraph and added this reference.

This manuscript is a resubmission of an earlier submission. The following is a list of the peer review reports and author responses from that submission.

Round 1

Reviewer 1 Report

The authors examined the usefulness of NC-AMRI in the surveillance of HCC. Since surveillance is a long-term, periodic process, cost-effectiveness is a very important issue, and it is important to explore methods beyond the currently recommended periodic US. Unfortunately, I believe that the study does not sufficiently demonstrate the superiority of NC-AMRI over other imaging modalities. The details are as follows.

[Major points]

The authors pointed out that the cost and time of CE-MRI versus NE-AMRI is superior to NE-AMRI, but I think this does not mean that NE-AMRI is superior because they have different tumor detection capabilities and serve different purposes. I understood that the patients in this study were at high risk for hepatocarcinogenesis, and if this is the case, the advantage of NE-AMRI over CE-MRI in clinical surveillance must be clearly demonstrated. In the high-risk group for hepatocarcinogenesis, a detailed examination for HCC (such as CE-MRI and CE-CT) will be necessary, and the authors need to explain what they consider the advantages of NE-AMRI for such patients. If the NE-AMRI as described in this study were used for surveillance, the Reader 2 results indicate that approximately 20% of HCCs would have been missed if the readers were not well trained. To compensate for this shortcoming, the need for double reading has been mentioned in several parts, but if double reading is also implemented, the number of readers per image will have to be doubled, and skilled readers will have to review all images, which will further increase the cost and time of readers. This may cancel out the cost-saving advantage of NR-AMRI. This would not lead to the implementation of NE-AMRI for patients in the high-risk group for hepatocarcinogenesis, and I think that periodic CE-MRI and CE-CT would be the correct surveillance.

Although the manuscript describes only briefly about the usefulness of MRI compared to CT, it would be better to explain what the authors think about it. To show the benefit of this study, it is necessary to show how NC-AMRI is beneficial not only in comparison with US but also in comparison with CT. If radiation exposure is an issue, the authors should not stop at the word "generally" (line 69), but they need to show what the adverse effects of repeated CT examinations every few months would be on the patient. If the patient is not at high risk for carcinogenesis, such as Lynch's syndrome, I would think that exposure to radiation once every 3-4 months would not be a problem, but what do authors think?

The sensitivity of 80-98% and the specificity of 72-91% are good results, but further discussion is needed to clarify the superiority of NC-AMRI over previously reported methods such as US and CT. As the authors pointed out, I agree that the advantage of US is its simplicity and high detectability relative to its cost. The explanation in lines 305-308 is unclear about the advantage of MRI. What is the advantage of this study versus ref 22? If the benefit of this study is to be demonstrated, it is necessary to explain in detail how the cost or detectability of NC-AMRI are superior to US, as the time required for NC-AMRI is likely to be inferior to that of US. Even if a direct comparison with US is not possible within this study, a comparison with known findings is necessary.

I felt that this study was poorly planned and carried out, since about 30% (lines 359-387) of the discussions is devoted to explaining the limitations. Lines 388-396 are also excuses and do not directly relate to the results of this study. The discussion is not a part to excuse for the results of the study, but about how the findings of the study will benefit current and future medicine.

[Minor points]

In the manuscript, French commas are mainly used for decimal points, but British periods are used in some places. In particular, the details of the MRI sequence in lines 121-134 are mixed up and very difficult to understand. Especially in this section, I could not understand how it was separated because of the mixture of separating commas and decimal point commas. I think that the overall appearance should be organized regarding the description of decimal points and separators.

Some of the abbreviations are duplicated. And the abbreviation “CT” is used for just a few times. The authors should organize the entire document regarding the use of abbreviations.

Reviewer 2 Report

To the authors:

“Potential of a non-contrast-enhanced abbreviated MRI screening protocol (NC-AMRI) in high-risk patients under surveillance for HCC”

My comments are as follows.

  1. How many patients with tattoos in this study?
  2. Please show the patient characteristics both with HCC and without HCC in another Table.
  3. How often do you recommend to evaluate with non-contrast-enhanced abbreviated MRI (NC-AMRI) for detecting HCC in high-risk patients?
  4. How much does it cost per NC-AMRI in your country?
  5. Also, how much does the patient pay per NC-AMRI in your country?

I hope that my comments will be useful in improving the article.

Reviewer 3 Report

in manuscript cancers-1594825, the authors retrospectively assessed the diagnostic performance of an abbreviated non-contrast MRI protocol for surveillance of hepatocellular carcinoma (HCC). The authors retrospectively included 225 patients (36 with HCC) over a 9 years period. All the patients had a contrast-enhanced MRI: the authors simulated the abbreviated MRI protocol by extracting from the examination the T2w sequences with fat saturation, the DWI sequences, and the T1-weighted in- and out-of-phase. the gold standard was the complete, contrast-enhanced examination. The images were reviewed by three readers with different experiences. the authors reported overall acceptable diagnostic accuracy of their abbreviated protocol.

Moreover, the authors also performed a double-reading session with improved diagnostic performance.   The quality of the manuscript is globally acceptable. however, there are some conceptual issues.

Introduction. some sentences should be rewritten. in particular, the authors should provide the background evidence supporting their studies without stating their impression or the rationale of the imaging protocol (which should be moved on material and methods). Moreover, the authors focus on the diagnosis of early HCC as the supposed target of surveillance. This can be misleading since they do non have a histopathological reference standard in their study nor this is an explicit target of surveillance programs. Please correct and improve.  

Materials and methods. please improve consistency. the authors state that the three readers considered the image quality, these results are not reported in results.  

Results and discussion: the results are interesting, in particular in terms of false-positive cases. Even if the diagnostic performance of MRI is superior to the of US, the results provided confirm the issues regarding the surveillance with CT or MRI (i.e. false positive rates). The authors in the discussion state that for screening programs a high sensitivity is preferable, supporting their results. However, this can be misleading in a surveillance/screening program since a high specificity is advisable, as also suggested by the EASL guidelines (reference 4). Even if the results can be interesting, their interpretation must be improved in particular when referring to the clinical setting of surveillance.